# A Little Selection Goes A Long Way! Parameter Efficient Domain Adaptive Object Detection via Noise-Guided Layer Selection

## Abstract

Domain Adaptive Object Detection (DAOD) aims to adapt a detector trained on a labeled source domain so that it generalizes well to a target domain with a different data distribution. Existing DAOD methods often fine-tune the entire source model on the target domain, which leads to parameter inefficiency and limits practical deployment on edge devices. In this paper, we demonstrate that fine-tuning only a subset of layers within the backbone can achieve comparable or even better performance. We propose **N**oise-**G**uided **L**ayer **S**election, **NGLS**, a method to identify backbone layers that best support learning domain-invariant representations. NGLS perturbs an auxiliary dataset with Gaussian noise, measures the cosine similarity of features across layers, and selects those layers whose similarity over the threshold. To demonstrate the effectiveness of our method, we integrate NGLS into two distinct DAOD tasks, Source-Free Object Detection (SFOD) and Unsupervised Domain Adaptive Object Detection (UDAOD). To further validate the generality of our method, we evaluate NGLS with two widely used detectors, Faster R-CNN (FRCNN) and Deformable DETR (DeDETR). The experimental results demonstrate that our method significantly reduces the number of required trainable parameters during adaptation while maintaining comparable or even surpassing performance compared to baseline methods. Specifically, in the Cityscapes to Foggy Cityscapes adaptation, we improve the performance of a DeDETR-based SFOD method by 0.8% mAP while reducing 98% of the model's trainable parameters, and we improve the performance of an FRCNN-based SFOD method by 2.1% mAP while reducing 93% of the trainable parameters.

## 1 Introduction

Object detection models often suffer significant performance drops when deployed across domains due to the domain gap between source training data and target testing data. For example, an object detection model pre-trained on clear-weather images may struggle to localize and classify objects under adverse weather conditions, such as fog or heavy rain. To address this issue, Domain Adaptive Object Detection (DAOD) is a crucial task. However, existing SFOD (VS et al. (2023); Liu et al. (2023); Khanh et al. (2024); Li et al. (2022a)) and UDAOD (Cao et al. (2023); Li et al. (2022b); Kennerley et al. (2024); Huang et al. (2024)) methods usually fine-tune the entire model on the target domain. These methods require updating a large number of parameters, resulting in low parameter efficiency and limiting their practical deployment on resource-constrained

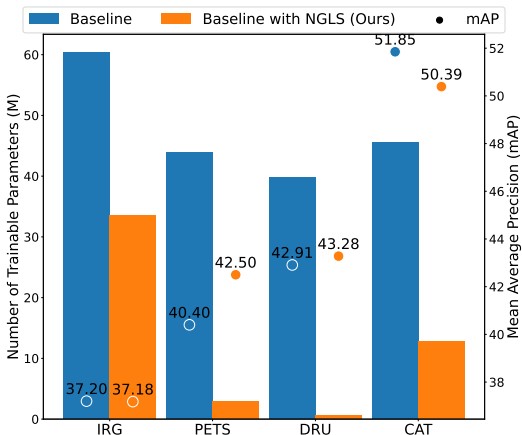

Figure 1: **Effectiveness of Noise-Guided Layer Selection (NGLS).** Integrating NGLS reduces the number of parameters, while maintaining or even improving their performance.

edge devices. Moreover, storing separate models for different target domains imposes a substantial burden on devices. Since the storage cost increases linearly with the number of domains, and the real world contains vast and diverse domains with potentially dynamic shifts, fine-tuning and maintaining a dedicated model for each domain is impractical. To overcome these limitations, we propose **N**oise-**G**uided **L**ayer **S**election (**NGLS**), a plug-and-play layer selection method that can be integrated into existing DAOD methods. As shown in Figure 1, NGLS significantly reduces the number of the required trainable parameters while maintaining, or even improving, the performance of the DAOD methods.

Recently, Meng et al. introduce Gaussian noise to analyze how autoregressive transformer language models store knowledge and recall factual associations. Building on these findings, Zhang et al. Zhang et al. (2024) examine how individual MLP layers within transformer blocks contribute to output predictions. Instead of updating all parameters in the foundation model, they selectively update a sparse set of task-relevant parameters, preserving the model's original capabilities while enabling it to continually acquire new knowledge. Inspired by these approaches Meng et al. (2022); Zhang et al. (2024), we aim to identify the layers within the detector that most significantly influence adaptation performance while preserving domain-invariance of the representation. Specifically, NGLS identifies backbone layers that contribute most to domain-invariant representation learning before adaptation. To achieve this, we inject Gaussian noise into the features of auxiliary images and measure the similarity between each backbone layer's outputs for clean and noise-injected inputs. Layers that can produce similar features under both conditions are considered more robust to domain shift. Notably, our method only requires a small amount of data (approximately 10–15 images) from an auxiliary dataset independent of both the source and target domains, yet effectively identifies the layers most critical for learning domain-invariant representations. By fine-tuning only these selected layers, NGLS achieves performance comparable to fine-tuning the entire model. We apply NGLS to several state-of-the-art DAOD methods to demonstrate its effectiveness and generality. Specifically, we integrate NGLS into two DAOD settings, SFOD and UDAOD, and evaluate it on different detectors, including Faster R-CNN and Deformable DETR.

Our main contributions are summarized below:

- We propose a novel plug-and-play layer selection method, NGLS, that leverages Gaussian noise perturbation to identify backbone layers most robust to domain shifts.
- By fine-tuning only the NGLS-selected layers, our approach substantially reduces the number of parameters while achieving comparable or superior performance to full-model fine-tuning in both SFOD and UDAOD settings.
- NGLS only requires a handful of unlabeled auxiliary images for robust layer selection, eliminating the need for source or target domain labels and supporting practical, data-efficient adaptation.

## 2 RELATED WORK

### 2.1 SELECTION OF DOMAIN-INVARIANT AND TASK-RELEVANT PARAMETERS

Recent studies (Meng et al. (2022); Zhang et al. (2024)) have focused on identifying task-relevant parameters in foundation models, updating only a small subset of parameters to retain the capabilities of the original model while enabling continual learning. Meng et al. introduce Gaussian noise to investigate how autoregressive transformer language models store and retrieve factual knowledge. Building on this, Zhang et al. analyze the contributions of individual MLP layers within transformer blocks to output predictions, proposing a sparse update strategy that targets only the most relevant parameters. In the context of domain adaptation, some studies also explore parameters or blocks that are most influential for adaptation. In Domain-Invariant Parameter Exploring (DIPE) (Wang et al. (2022)), the authors provide an important insight: *rather than attempting to learn domain-invariant representations, it is more effective to explore the domain-invariant parameters of the model.* To this end, they design a domain-balanced identifying criterion that examines whether parameters play consistent positive or negative roles in the same positions across the source and target models during forward propagation. To further enhance model performance during Test-Time Adaptation (TTA), Yu et al. propose Pseudo-Labeling for Online Test-Time adaptation (DPLOT) (Yu et al. (2024)) to identify specific blocks in a pre-trained network by comparing prototype features before

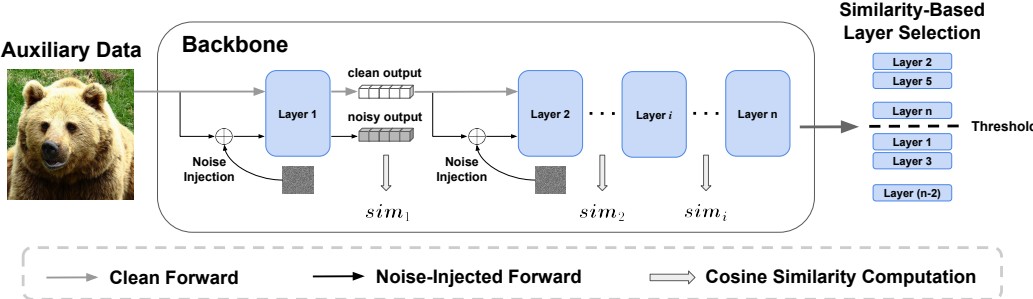

Figure 2: **Overview.** We provide the overview of **NGLS** at here. We utilize auxiliary data to select layers. *Clean Forward* denotes passing clean features into the layer without noise injection, while *Noise-Injected Forward* denotes injecting Gaussian noise into the clean features before feeding them into the layer. At each layer, we obtain both clean and noisy outputs corresponding to the clean and noise-injected inputs, and compute the cosine similarity between them. After obtaining similarity from each backbone layer, we select layers with high similarity for adaptation.

and after adapting each block with noisy images. Specifically, they compute the similarity between the original prototypes and those obtained after updating each block's parameters through entropy minimization with noisy inputs.

## 2.2 DOMAIN ADAPTIVE OBJECT DETECTION

Since collecting labeled data is challenging, many DAOD methods leverage unlabeled data to improve model performance, such as Source-Free Object Detection (SFOD) and Unsupervised Domain Adaptive Object Detection (UDAOD). The key difference between SFOD and UDAOD lies in whether labeled source data is used during fine-tuning. Existing SFOD (Chu et al. (2023); Li et al. (2021); VS et al. (2023); Liu et al. (2023); Khanh et al. (2024); Li et al. (2022a)) and UDAOD (Cao et al. (2023); Li et al. (2022b); Kennerley et al. (2024); Huang et al. (2024)) methods commonly employ Faster R-CNN (Ren et al. (2015)) or Deformable DETR (DeDETR) (Zhu et al. (2020)) as the base detector and adopt the Mean Teacher (MT) (Tarvainen & Valpola (2017)) framework as the training paradigm, where the student model is guided by the teacher model, and the teacher model is updated via exponential moving average (EMA). Among these methods, the Instance Relation Graph (IRG) (VS et al. (2023)) and Periodically Exchange Teacher-Student (PETS) (Liu et al. (2023)) are state-of-the-art (SOTA) SFOD approaches that employ different backbones in Faster R-CNN. Specifically, VS et al. introduce the instance relation graph network and a graph convolution network to enhance target domain representation, improving pseudo-label quality and boosting adaptation performance. Liu et al. propose PETS, which employs two teacher models to stabilize training, ensuring more reliable adaptation and improving adaptation results. Additionally, Khanh et al. propose Dynamic Retraining-Updating (DRU) (Khanh et al. (2024)) to address degradation in the mean teacher framework and is the first study to investigate the effectiveness of Deformable DETR (DeDETR) for SFOD. Their approach achieves SOTA performance compared to prior SFOD methods. In UDAOD, labeled source data and unlabeled target data are available during fine-tuning (adaptaiton) stage, several methods (Cao et al. (2023); Kennerley et al. (2024); Huang et al. (2024)) mitigate domain discrepancies by aligning image features across source and target domains through adversarial training. To further improve pseudo-label quality, Kennerley et al. propose Class-Aware Teacher (CAT) (Kennerley et al. (2024)) to reduce the class bias within the model, and achieve SOTA performance in UDAOD task. Although these methods effectively reduce the domain gap between the source and target domain, they typically do so by fine-tuning the entire source-pretrained model on the target domain, which requires updating a large number of parameters. Compared to these methods, NGLS significantly reduce the number of fine-tuned parameters while achieving competitive or even better performance.

## 3 METHODOLOGY

**Overview.** Several studies (Chen et al. (2019); Gao et al. (2019); Liu et al. (2020)) have highlighted the critical role of the backbone in many computer vision tasks, including object detection and image classification. The performance of these tasks largely depends on the quality of features extracted by the backbone. Moreover, in the context of Domain Adaptive Object Detection (DAOD), it is particularly important to generate domain-invariant representations. Motivated by these findings, we focus on selecting the backbone layers that contribute most to generating domain-invariant representations for DAOD.

Based on the findings in (Meng et al. (2022); Zhang et al. (2024)). We leverage Gaussian noise to identify backbone layers that best capture domain-invariant representations. Specifically, as illustrated in Figure 2, we inject Gaussian noise into the clean input at each layer, where the clean input is the clean output of the previous layer, to generate the noise-injected input. Both the clean and noisy inputs are passed through the layer to produce corresponding outputs, and we compute their cosine similarity to evaluate whether the layer remains stable under domain shift. For example, at layer $i$, we inject Gaussian noise into the clean output of layer $(i-1)$ and feed both the clean output $(i-1)$ and the noise-injected one into layer $i$. We then compute the cosine similarity between the noisy and clean outputs at layer $i$ to evaluate whether the layer produces consistent features under domain shifts. This process is repeated for each subsequent layer to obtain similarity scores across the backbone. After obtaining similarity scores from every layer within the backbone, we select layers that maintain high similarity for adaptation.

### 3.1 THE ROLE OF BACKBONE IN DOMAIN ADAPTIVE OBJECT DETECTION

To demonstrate that the backbone plays a crucial role in DAOD, we conduct a simple experiment using our baseline DAOD methods. Specifically, we freeze different components of the detector to observe which one has the greatest influence on model performance during fine-tuning. For example, in Periodically Exchange Teacher-Student (PETS) (Liu et al. (2023)), the base model is Faster R-CNN (FRCNN), which consists of a backbone, a region proposal network (RPN), and a region of interest (ROI) head. We freeze each component of Faster R-CNN in turn and follow the PETS fine-tuning pipeline to adapt the model from source to target domain. We perform four experiments: freezing the backbone, freezing the RPN, freezing the ROI head, and freezing both the RPN and ROI head, and evaluate which setting leads to the most significant performance degradation in the target domain. As an example, we consider the case where the backbone is frozen, which can be formulated as:

$$(\theta_s^{rpn}; \theta_s^{roi}) \leftarrow \nabla \mathcal{L}_{detection}(x_t, y_{pseudo}), \tag{1}$$

where $x_t \in D_t$ is target domain data, $\theta_s$ is the source pre-trained weight, and $y_{pseudo}$ is pseudo-label. Note that since PETS is an SFOD method, only target domain data is available during adaptation. The same analysis is applied to Dynamic Retraining-Updating (DRU) Khanh et al. (2024), which uses Deformable DETR (DeDETR) as its base model. We divide DeDETR into three components, the backbone, the encoder, and the decoder, and repeat the same procedure as with FRCNN. As shown in Figure 3, we conduct experiments on Cityscapes to Foggy Cityscapes, where freezing the backbone causes a substantial performance drop in all baseline methods, while freezing other components results in only minor differences compared to full-model fine-tuning. This indicates that the backbone plays the most critical role in the model's adaptation to a new, unseen domain.

### 3.2 NOISE-GUIDED LAYER SELECTION

To identify the layers responsible for generating domain-invariant representations, we hypothesize that these layers produce similar outputs when processing inputs with domain shifts. As discussed in Section 3.1, the backbone has the greatest influence on adaptation performance. Therefore, we apply our analysis to the backbone to select layers that contribute most to generating domain-invariant representations. As illustrated in Figure 2, we feed both the clean input and a noise-injected input into each backbone layer and compute the cosine similarity between the corresponding outputs. This process can be formulated as:

$$v_i^{clean} = f^{[i]}(v_{i-1}^{clean}), \tag{2}$$

$$v_i^{noisy} = f^{[i]}(v_{i-1}^{clean} + \epsilon), \text{where } \epsilon \sim \mathcal{N}(m, \sigma^2), \tag{3}$$

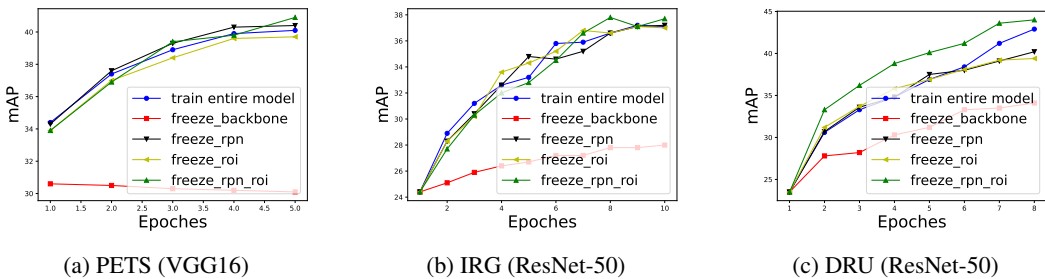

| (a) PETS (VGG16) | (b) IRG (ResNet-50) | (c) DRU (ResNet-50) |

Figure 3: **Importance of Backbone**. The results demonstrate that across different backbones (VGG16 and ResNet-50) and detectors (Faster R-CNN with IRG and PETS, and Deformable DETR with DRU), the lowest performance occurs when the backbone is frozen during adaptation. This experiment is conducted on the Cityscapes to Foggy Cityscapes adaptation.

$$sim_i = cos(v_i^{clean}, v_i^{noisy}), \qquad (4)$$

where $v_i^{clean}$ is the output feature of the $i^{th}$ backbone layer $f^{[i]}$ when fed with clean input (e.g., the clean output $v_{i-1}^{clean}$ from the previous layer $f^{[i-1]}$), $v_i^{noisy}$ is the output feature of $f^{[i]}$ when fed with noise-injected input ($v_{i-1}^{clean}+\epsilon$), and $sim_i$ is the cosine similarity between outpus from $i^{th}$ backbone layer, $v_i^{clean}$ and $v_i^{noisy}$, respectively. After obtaining the similarity scores for each backbone layer $\{sim_1,...,sim_i,...,sim_n\}$, we select the layers whose similarity exceeds a predefined threshold and fine-tune only these layers following the baseline DAOD methods.

## 4 EXPERIMENTS

**Datasets and Evaluation.** To demonstrate the effectiveness of our selection method. We evaluate the performance across various domain shifts, including (1) Cityscapes to Foggy Cityscapes (C2F), (2) Cityscapes to BDD100k (C2B), (3) Sim10k to Cityscapes Car (S2C), (4) PASCAL VOC to Clipart (P2C), (5) PASCAL VOC to Watercolor (P2W), and (6) Cityscapes to Dusk Rainy (C2D). **Cityscapes** (Cordts et al. (2016)) is a real urban scene dataset with 2,975 training images and 500 validation images, containing eight classes: person, rider, car, train, bicycle, motorbike, truck, and bus. **Foggy Cityscapes** (Sakaridis et al. (2018)) is a synthetic dataset derived from Cityscapes, simulating three fog levels (0.02, 0.01, and 0.005) to represent varying visibility conditions. **BDD100k** (Yu et al. (1805)) is a large-scale driving dataset, and its daytime subset is selected as the target domain. The training and validation sets contain 70,000 and 10,000 images, respectively. **Sim10k** (Johnson-Roberson et al. (2016)) is a synthetic dataset rendered from the video game Grand Theft Auto V (GTA V), containing 10,000 images of cars. **Cityscapes Car** retains only car images from the Cityscapes dataset, discarding all other categories. **PASCAL VOC** (Everingham et al. (2010)) contains 20 categories of common objects from real world with bounding box and class annotations. **Clipart1k** (Inoue et al. (2018)) contains clipart images and shares the same 20 classes with PASCAL VOC dataset. **Watercolor2k** (Inoue et al. (2018)) contains watercolor style images, which consists of images from 6 classes and shares with the same classes in PASCAL VOC dataset. **Dusk Rainy** (Wu et al. (2021)) contains 3,501 rainy images selected from the BDD100k dataset. Following prior works, we report AP50 as the mean average precision (mAP) to evaluate detection performance.

We consider four state-of-the-art (SOTA) DAOD methods: Instance Relation Graph (IRG) (VS et al. (2023)), Periodically Exchange Teacher-Student (PETS) (Liu et al. (2023)), Dynamic Retraining-Updating (DRU) (Khanh et al. (2024)), and Class-Aware Teacher (CAT) (Kennerley et al. (2024)) as our baselines. We apply NGLS to these methods for evaluation.

**Implementation Details.** Following the baseline methods, our approach is implemented using PyTorch. IRG, PETS, and CAT employ Faster R-CNN (Ren et al. (2015)) as their base model, with IRG using ResNet-50 as the backbone and PETS and CAT using VGG16. In contrast, DRU uses DeDETR (Zhu et al. (2020)) as the base model with ResNet-50 as the backbone. The GPU memory usage of the baseline method is as follows: IRG (ResNet-50) = 810.43 MB, PETS (VGG16) = 693.54 MB, DRU (ResNet-50) = 1192.79 MB, and CAT (VGG16) = 713 MB. We use COCO

| | Method | Detector | Params (M) | GPU Memory | person | rider | car | truck | bus | train | motor | bicycle | mAP |
|---|---|---|---|---|---|---|---|---|---|---|---|---|---|
| | AT | FRCNN | 45.5 | 0 | 45.5 | 55.1 | 64.2 | 35 | 56.3 | 54.3 | 38.5 | 51.9 | 50.9 |
| | CMT | FRCNN | 45.5 | 0 | 45.9 | 55.7 | 63.7 | 39.6 | 66 | 38.8 | 41.4 | 51.2 | 50.3 |
| UDAOD | CAT | FRCNN | 45.5 | 0 | 44.6 | 57.1 | 63.7 | 40.8 | 66 | 49.7 | 44.9 | 53 | **52.5** |
| | CAT* (Baseline) | FRCNN | 45.5 | 0 | 45.9 | 57.2 | 64.3 | 38.9 | 63.2 | 50.5 | 40.5 | 53.8 | 51.8 |
| | CAT (Ours) | FRCNN | **12.8** | -18% | 44.2 | 55.5 | 62.3 | 38.5 | 56.9 | 51.9 | 40.3 | 53.1 | 50.4 |
| | DRU | DeDETR | 39.8 | 0 | 48.3 | 51.5 | 62.5 | 26.2 | 43.2 | 34.1 | 34.2 | 48.6 | 43.6 |
| | DRU* (Baseline) | DeDETR | 39.8 | 0 | 47.8 | 50.7 | 63.3 | 24.6 | 40.1 | 34.9 | 34.5 | 47.6 | 42.9 |
| | DRU (Ours) | DeDETR | **0.58** | -20% | 48 | 49.2 | 64.8 | 26.8 | 42.8 | 38.5 | 35 | 44.5 | **43.7** |
| SFOD | IRG | FRCNN | 60.3 | 0 | 37.4 | 45.2 | 51.9 | 24.4 | 39.6 | 25.2 | 31.5 | 41.6 | 37.1 |
| | IRG* (Baseline) | FRCNN | 60.3 | 0 | 35.9 | 44.2 | 51.5 | 24.1 | 41 | 34.9 | 29 | 40.3 | 37.2 |
| | IRG (Ours) | FRCNN | 33.5 | -22% | 32.7 | 43.9 | 51.2 | 26.9 | 41.1 | 37.2 | 26.8 | 37.2 | 37.1 |
| | PETS | FRCNN | 43.8 | 0 | 46.1 | 52.8 | 63.4 | 21.8 | 46.7 | 5.5 | 37.4 | 48.4 | 40.3 |
| | PETS* (Baseline) | FRCNN | 43.8 | 0 | 45.9 | 52.4 | 63.4 | 19.9 | 47.8 | 7.2 | 36.9 | 47.5 | 40.1 |
| | PETS (Ours) | FRCNN | 2.9 | -23% | 45.9 | 52.8 | 63.4 | 22.8 | 47.2 | 23.4 | 34.5 | 47.4 | 42.2 |

Table 1: **Cityscapes to Foggy Cityscapes.** We present the performance of baseline methods along-side the results obtained after integrating our approach. * indicates performance reproduced using the officially released code, while "Ours" denotes results achieved by applying our method. "FR-CNN" refers to Faster R-CNN, and "DeDETR" refers to Deformable DETR. "Params" indicates trainable parameters during adaptation. "GPU Memory" indicates GPU memory usage, where "-x%" denotes a reduction of x% GPU memory usage compared to full-model fine-tuning. The best mAP and the lowest number of trainable parameters are highlighted in **bold**.

| | Methods | Detector | Params (M) | GPU Memory | person | rider | car | truck | bus | motor | bicycle | mAP |
|---|---|---|---|---|---|---|---|---|---|---|---|---|---|
| | SED | FRCNN | 43.8 | 0 | 32.4 | 32.6 | 50.4 | 20.6 | 23.4 | 18.9 | 25 | 29 |
| | $A^2$SFOD | FRCNN | 43.8 | 0 | 33.2 | 36.3 | 50.2 | 26.6 | 24.4 | 22.5 | 28.2 | 31.6 |
| SFOD | PETS | FRCNN | 43.8 | 0 | 42.6 | 34.5 | 62.4 | 19.3 | 17 | 16.9 | 26.3 | **31.3** |
| | PETS* (Baseline) | FRCNN | 43.8 | 0 | 42.52 | 33.74 | 62.27 | 18.97 | 17.65 | 15.61 | 25.56 | 30.91 |
| | PETS (Ours) | FRCNN | **9.9** | -21% | 40.51 | 27.09 | 60.91 | 18.66 | 15.06 | 14.87 | 22.38 | 28.5 |

Table 2: **Cityscapes to BDD.** Results of adaptation from small-scale (Cityscapes) to large-scale (BDD100k) dataset (C2B). *: reproduced using official code. **Bold**: the best mAP and the lowest trainable parameters.

as auxiliary data, since this dataset is not included in any adaptation benchmark. We use Gaussian noise with a mean of 0.5 and a standard deviation of 5.0 in the layer selection process. The threshold for selecting layers is set to 0.9998 in IRG, 0.7 in DRU, 0.6 in CAT, and in PETS, it is 0.99 for Cityscapes to Foggy Cityscapes and 0.9 for Sim10k to Cityscapes Car. The hyperparameters used during fine-tuning follow those of the respective baseline methods. For a fair comparison, all baseline methods are reproduced using their officially released code without modifications. We apply our approach to the released code and use the same hyperparameters. All experiments are conducted on a single NVIDIA GeForce RTX 3090 GPU, except for CAT, which uses four GPUs.

## 4.1 Comparison with baseline methods

**C2F: adaptation from normal driving scene to foggy driving scene.** Table 1 presents the evaluation results of the detector pre-trained on Cityscapes and adapted to Foggy Cityscapes. The results show that integrating our method significantly reduces the number of trainable parameters ("Params") in the baseline methods while achieving comparable or even superior performance. Specifically, in CAT, we reduce 70% of the trainable parameters and 18% of GPU memory usage compared to the baseline, while maintaining competitive performance. In IRG, we reduce 44% of the trainable parameters and 20% of GPU memory usage, also achieving competitive performance. In PETS, we reduce 93% of the trainable parameters and 23% of GPU memory usage, surpassing the baseline by 2.1% mAP. In DRU, we reduce 98% of the trainable parameters and 20% of GPU memory usage while achieving state-of-the-art performance compared to previous methods.

**C2B: adaptation from small-scale to large-scale dataset.** To validate the ability to adapt from small-scale to large-scale datasets, existing methods evaluate their approaches using a detector pre-trained on Cityscapes and adapted to BDD100k. Following previous studies, we retain the eight BDD100k classes that align with those in Cityscapes. Because performance on the "train" category is consistently near zero, we follow PETS and report mAP scores only for the remaining seven categories. As shown in Table 2, the results demonstrate that our method achieves competitive

| | Method | Detector | Params (M) | GPU Memory | bike | bird | car | cat | dog | prsn | map |
|---|---|---|---|---|---|---|---|---|---|---|---|
| SFOD | SED | FRCNN | 45 | 0 | 76.2 | 44.9 | 49.3 | 31.6 | 30.6 | 55.2 | 47.9 |
| | Mean Teacher | FRCNN | 43.8 | 0 | 73.6 | 47.6 | 46.6 | 28.5 | 29.4 | 56.6 | 47.1 |
| | IRG | FRCNN | 60.3 | 0 | 75.9 | 52.5 | 50.8 | 30.8 | 38.7 | 69.2 | **53** |
| | IRG* (Baseline) | FRCNN | 60.3 | 0 | 75.9 | 53.6 | 48.5 | 30.1 | 36.3 | 62 | 51 |
| | IRG (Ours) | FRCNN | **33.5** | -22% | 79.7 | 49.8 | 49.2 | 30.7 | 33.9 | 63.6 | 51.2 |

Table 3: **PASCAL VOC to Watercolor.** Results of adaptation from realistic (PASCAL VOC) to artistic (Watercolor) images (P2W). *: reproduced using official code. **Bold**: the best mAP and the lowest trainable parameters.

| | Method | Detector | Params (M) | GPU Memory | aero | bcycle | bird | boat | bottle | bus | car | cat | chair | cow | table | dog | horse | bike | prsn | plnt | sheep | sofa | train | tv | mAP |
|---|---|---|---|---|---|---|---|---|---|---|---|---|---|---|---|---|---|---|---|---|---|---|---|---|---|
| SFOD | SED | FRCNN | 45.5 | 0 | 20.1 | 51.5 | 26.8 | 23 | 24.8 | 64.1 | 37.6 | 10.3 | 36.3 | 20 | 18.7 | 13.5 | 26.5 | 49.1 | 37.1 | 32.1 | 10.1 | 17.6 | 42.6 | 30 | 29.5 |
| | Mean Teacher | FRCNN | 43.8 | 0 | 22.3 | 47.3 | 27.3 | 19.7 | 30.5 | 54.2 | 36.2 | 10.3 | 35.1 | 20.6 | 20.2 | 12.3 | 28.7 | 53.1 | 47.5 | 42.4 | 9.09 | 21.1 | 42.3 | 50.3 | 31.5 |
| | IRG | FRCNN | 60.3 | 0 | 20.3 | 47.3 | 27.3 | 19.7 | 30.5 | 54.2 | 36.2 | 10.3 | 35.1 | 20.6 | 20.2 | 12.3 | 28.7 | 53.1 | 47.5 | 42.4 | 9.09 | 21.1 | 42.3 | 50.3 | 31.5 |
| | IRG* (Baseline) | FRCNN | 60.3 | 0 | 21.1 | 54.8 | 26.8 | 20.8 | 33.9 | 56.4 | 33.9 | 9 | 39.2 | 14.3 | 25.4 | 3.4 | 36.9 | 50.8 | 48 | 42.8 | 16.3 | 23.1 | 37.4 | 39.1 | **31.7** |
| | IRG (Ours) | FRCNN | **33.5** | -22% | 24.5 | 51.4 | 25.2 | 19.6 | 30.3 | 54 | 35 | 9 | 35.2 | 6.2 | 21.6 | 4.9 | 31.2 | 51.5 | 49.1 | 43.4 | 18.1 | 14.3 | 37.1 | 40.8 | 30.1 |

Table 4: **PASCAL VOC to Clipart.** Results of adaptation from realistic (PASCAL VOC) to artistic (Clipart) images (P2C). *: reproduced using official code. **Bold**: the best mAP and the lowest trainable parameters.

| | Methods | Detector | Params (M) | GPU Memory | person | rider | car | truck | bus | motor | bicycle | mAP |
|---|---|---|---|---|---|---|---|---|---|---|---|---|
| SFOD | IRG* (Baseline) | FRCNN | 60.3 | 0 | 22.28 | 15.91 | 41.8 | 16.96 | 22.65 | 1.07 | 14.7 | 19.34 |
| | IRG (Ours) | FRCNN | 33.5 | -22% | 22.82 | 15.91 | 45.22 | 15.33 | 20.4 | 4.07 | 13.06 | 19.54 |

Table 5: **Cityscapes to Dusk Rainy.** Results for adaptation from the normal driving scene (Cityscapes) to the rainy driving (Dusk Rainy) scene (C2R). *: reproduced using official code. **Bold**: the best mAP and the lowest trainable parameters.

| | Method | Detector | Params (M) | GPU Memory | AP car |
|---|---|---|---|---|---|
| SFOD | DRU | DeDETR | 39.8 | 0 | **58.7** |
| | DRU* (Baseline) | DeDETR | 39.8 | 0 | **58.7** |
| | DRU (Ours) | DeDETR | **2.9** | -18% | 57.2 |
| | IRG | FRCNN | 60.3 | 0 | 45.2 |
| | IRG* (Baseline) | FRCNN | 60.3 | 0 | 46 |
| | IRG (Ours) | FRCNN | 33.5 | -22% | 47.3 |
| | PETS | FRCNN | 43.8 | 0 | 57.8 |
| | PETS* (Baseline) | FRCNN | 43.8 | 0 | **57.9** |
| | PETS (Ours) | FRCNN | 9.9 | -21% | 56 |

Table 6: **Sim10k to Cityscapes Car.** Results of adaptation from synthetic (Sim10K) to real (cityscapes Car) scenes (S2C). *: reproduced using official code. **Bold**: the best mAP and the lowest trainable parameters.

performance in the target domain compared to the baseline, while reducing the number of trainable parameters by 77% and GPU memory usage by 21% in PETS.

**P2W: adaptation from realistic to artistic.** To demonstrate adaptation effectiveness, existing studies commonly evaluate their methods on benchmarks with a large domain gap between the source and target domains. A typical setting uses PASCAL VOC as the source domain and Watercolor as the target domain. In this setting, we investigate whether our method can maintain competitive performance with fewer trainable parameters under the domain shift from real to artistic images. As shown in Table 3, applying our method to IRG yields nearly identical results while reducing the number of trainable parameters by 44% and GPU memory usage by 22%.

**P2C: adaptation from realistic to artistic.** Similar to P2W, which evaluates adaptation from real to artistic domains, existing methods use PASCAL VOC as the source domain and Clipart as the target domain. As shown in Table 4, applying our method to IRG achieves comparable results while reducing the number of trainable parameters by 44% and GPU memory usage by 22%.

**C2R: adaptation from normal driving scene to rainy driving scene.** To evaluate the performance of our method under different weather conditions, we further conduct an experiment adapting the

model from Cityscapes to the Dusk Rainy dataset. Since existing DAOD methods do not report results for this adaptation setting, and the performance of PETS is poor (approximately 5 mAP in the target domain), we only present results for IRG. As shown in Table 5, integrating our method into IRG achieves comparable performance while reducing the number of trainable parameters by 44% and GPU memory usage by 22%.

**S2C: adaptation from synthetic to real scenarios.** Data collection is often challenging, making synthetic data a valuable alternative. Existing methods adapt models pre-trained on synthetic data to real-world scenes to evaluate their adaptation capabilities. Specifically, they use Sim10k as the source domain and adapt to Cityscapes Car. As shown in Table 6, our method achieves competitive performance compared to baseline approaches while significantly reducing trainable parameters by 44% in IRG, 77% in PETS, and 92.7% in DRU, and GPU memory usage by 22% in IRG, 21% in PETS, and 18% in DRU.

### 4.2 ABLATION STUDY

**Different Threshold for Selecting Layers.** As mentioned in Section 3.2, after computing the similarity for all backbone layers, we apply a threshold to select the layers used for adaptation. This demonstrates that adapting only the layers with high similarity (e.g., fewer trainable parameters) can achieve performance comparable to using more layers. In Table 7, we report the results obtained with different thresholds for layer selection. This ablation study,

| Method | Threshold | Params (M) | mAP |
|---|---|---|---|
| IRG | 1.0 | 33.1 | 36.1 |
| | 0.9998 | 33.56 | 37.1 |
| | 0.99 | 34.5 | 37 |
| | Entire Backbone | 52 | 37.5 |
| | Entire Model (Baseline) | 60.3 | 37.2 |
| PETS | 0.99 | 2.9 | 42.5 |
| | 0.9 | 9.9 | 41.8 |
| | Entire Backbone | 14.7 | 41.9 |
| | Entire Model (Baseline) | 43.8 | 40.4 |
| DRU | 0.9 | 0.2 | 38.8 |
| | 0.7 | 0.5 | 43.2 |
| | 0.6 | 0.8 | 43.7 |
| | 0.3 | 2.9 | 42.5 |
| | Entire Backbone | 23.2 | 44 |
| | Entire Model (Baseline) | 39.8 | 42.9 |
| CAT | 0.7 | 10.4 | 49.4 |
| | 0.6 | 12.8 | 50.4 |
| | Entire Backbone | 15 | 50.8 |
| | entire model | 45.5 | 51.8 |

Table 7: **Different Threshold for Selecting Layers.** We present experimental results on the Cityscapes to Foggy Cityscapes adaptation, using different thresholds to select layers for fine-tuning on the target domain. "Entire Backbone" indicates fine-tuning only the backbone while keeping other modules frozen, and "Entire Model" denotes the baseline performance.

conducted on the Cityscapes to Foggy Cityscapes adaptation, shows that using a higher threshold (fewer trainable parameters) can yield comparable or even better performance than using a lower threshold.

**Different Data for Selecting Layers.** We conduct this ablation study on the Cityscapes to Foggy Cityscapes adaptation, where the model is pre-trained on labeled Cityscapes data and then adapted to Foggy Cityscapes. To investigate whether using random data for layer selection affects performance, we use three different auxiliary datasets as well as the source data (Cityscapes) to select layers. As shown in Table 8, the choice of data for layer selection does not significantly influence the selected layers.

**High-Similarity Layers vs. Low-Similarity Layers.** To demonstrate that selecting layers with high similarity benefits the model's adaptation to a new domain, we conduct an experiment comparing the performance of fine-tuning layers with low similarity versus high similarity to the target domain. This experiment is performed on the Cityscapes to Foggy Cityscapes adaptation, with mAP used to measure performance in the target domain. As shown in Ta-

| Method | Dataset | Params (M) | mAP |
|---|---|---|---|
| IRG | COCO | 33.5 | 37.1 |
| | Daytime Sunny | 32.4 | 37.4 |
| | CelebA | 33.5 | 37.3 |
| | Cityscapes | 33.1 | 37.1 |
| PETS | COCO | 2.9 | 42.5 |
| | Daytime Sunny | 2.9 | 42 |
| | CelebA | 2.9 | 42.3 |
| | Cityscapes | 2.9 | 42.3 |

Table 8: **Different Data for Selecting Layers.** We present experimental results on the Cityscapes to Foggy Cityscapes adaptation. "Dataset" refers to the dataset used for selecting layers. After selecting the layers, we fine-tune the selected layers on the target domain (Foggy Cityscapes). The results demonstrate that using either the Cityscapes dataset (source data) or auxiliary dataset (COCO, Daytime Sunny, and CelebA) for layer selection yields very similar results.

| Method | Selection Criterion | Params (M) | mAP |
|--------|---------------------|------------|-----|
| | High Similarity (above 0.9998) | 33.56 | 37.1 |
| IRG | Low Similarity (below 0.8) | 28.8 | 35 |
| | Entire Model (Baseline) | 60.3 | 37.2 |
| | High Similarity (above 0.99) | 2.9 | 42.5 |
| PETS | Low Similarity (below 0.5) | 14.2 | 36.8 |
| | Entire Model (Baseline) | 43.8 | 40.4 |
| | High Similarity (above 0.7) | 0.5 | 43.2 |
| DRU | Low Similarity (below 0.3) | 20.24 | 41.1 |
| | Entire Model (Baseline) | 39.8 | 42.9 |

Table 9: **High-Similarity Layers vs. Low-Similarity Layers.** We present experimental results on the Cityscapes to Foggy Cityscapes adaptation, comparing the performance of fine-tuning layers with low similarity versus layers with high similarity to the target domain.

ble 9, fine-tuning layers with high similarity not only achieves better performance than fine-tuning low-similarity layers but also requires fewer parameters.

## 5 LAYER ANALYSIS

In Figure 4, we present the similarity results for layer selection, as discussed in Section 3.2 of the main paper. The results show that the deeper layers of ResNet-50 produce similar features when processing inputs with a domain gap, whereas VGG16 exhibits this behavior in its shallower layers. These observations align with (Cadena et al. (2018)), which shows that early layers of VGG exhibit strong response invariance across domains, while ResNet's lower layers are less invariant. Additionally, prior work (Yu et al. (2024); Zhou et al. (2021); Choi et al. (2022); Wang et al. (2021)) suggests that domain-specific representation is primarily captured in the shallow layers of ResNet, further supporting our findings.

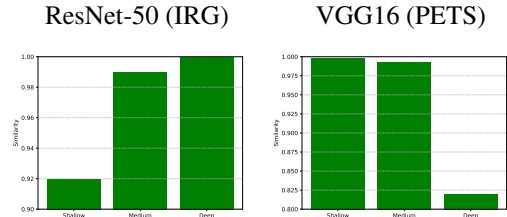

Figure 4: **Layer Analysis.** We present layer-wise cosine similarity results for different backbones (ResNet-50 and VGG16). Based on these results, we set a threshold to select the layers used for adaptation.

## 6 CONCLUSION

We propose Noise Guided Layer Selection (NGLS), a novel approach for source-free object detection that addresses the challenges of catastrophic forgetting and inefficiency when adapting to multiple domains. By selectively updating only the layers within the backbone that are most crucial for adaptation based on our analysis between source and pseudo domains, NGLS significantly reduces the number of training parameters, while maintaining competitive performance. This allows for rapid and efficient adaptation to target domains without the need for full model fine-tuning. Moreover, our experiments demonstrate that our method effectively adapts object detectors to target domains without compromising source domain performance.

**Limitaions.** The major limitation of our method is that it is constrained by the performance upper bound of adapting only the backbone to the target domain. Since fine-tuning only the backbone produces slightly lower results compared to fine-tuning the entire model in PETS, the performance on certain target domains in PETS, such as S2C and C2B, drops slightly after applying our method.

ETHICS STATEMENT

To the best of our knowledge, this work has no potential negative social impact. Our selection method, NGLS, has the potential to benefit various object detection tasks. Detectors fine-tuned on the target domain often suffer from catastrophic forgetting, losing knowledge acquired from previous domains. Our method effectively preserves the capabilities of the detector pre-trained on earlier domains, while still achieving competitive performance on the new domain.

REPRODUCIBILITY

To ensure reproducibility, we have provided sufficient implementation details. In addition, we will release our source code and model weights upon paper acceptance.

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

## A   USAGE OF LARGE LANGUAGE MODELS

The core method development in this research does not involve LLMs as any important, original, or non-standard components. We only use LLMs for polish writing.

