# OpenReview forum: "A Little Selection Goes A Long Way! Parameter Efficient Domain Adaptive Object Detection via Noise-Guided Layer Selection"
_ICLR.cc/2026/Conference — ICLR 2026 Conference Withdrawn Submission_

### Official Review · Reviewer_DzeF · 2025-10-24

**Soundness:** 3
**Presentation:** 2
**Contribution:** 2
**Rating:** 4
**Confidence:** 5

**Summary:**

This paper addresses the parameter inefficiency issue of existing Domain Adaptive Object Detection (DAOD) methods, which typically require fine-tuning the entire source model on the target domain, thereby limiting deployment on edge devices. The authors propose Noise-Guided Layer Selection (NGLS), a plug-and-play method that identifies backbone layers critical for learning domain-invariant representations by perturbing auxiliary data with Gaussian noise and measuring cosine similarity between clean and noisy layer outputs. NGLS is integrated into two DAOD tasks (Source-Free Object Detection (SFOD) and Unsupervised Domain Adaptive Object Detection (UDAOD)) and evaluated on two detectors (Faster R-CNN and Deformable DETR) across six domain shift scenarios. Key results include reducing trainable parameters by up to 98% while maintaining or improving performance (e.g., 2.1% mAP gain for FRCNN-based SFOD and 0.8% mAP gain for DeDETR-based SFOD in Cityscapes to Foggy Cityscapes adaptation).

**Strengths:**

1. NGLS introduces a simple yet effective noise-guided layer selection strategy that addresses a critical real-world challenge (parameter inefficiency). Its plug-and-play design allows seamless integration with existing DAOD methods, enhancing usability.
2. The paper tests NGLS across diverse domain shifts (weather, dataset scale, realism to art, synthetic to real) and multiple state-of-the-art baselines (IRG, PETS, DRU, CAT). This breadth demonstrates the method’s generality and robustness.
3. The authors conduct thorough ablations on threshold selection, auxiliary data choice, and high vs. low similarity layers, validating the core assumptions of NGLS and providing actionable insights for parameter tuning.

**Weaknesses:**

1. One major concern for this paper is the motivation of parameter-efficient model training for the SFOD problem. It feels like this paper simply borrows the philosophy of parameter-efficient fine-tuning in the large language model into SFOD. However, it is not the same case. I wonder if it is important to research parameter-efficient training for SFOD because the parameters for the detector are usually on a small scale compared with LLMs.
2. As shown in Table 6, the proposed method reduces the number of updated parameters, but the GPU memory only reduces by around 20%.
3. There are many ways to add perturbations for the feature; the authors need to compare different implementation ways, such as dropout.

**Questions:**

1.  In Table 9, the Selection Criterion is very weird and arbitrary. For example, IRG high similarity is defined as above 0.9998, while that of PETS is above 0.99. Why is there a difference between different methods to choose the threshold,d, and how to choose the threshold?
2. This paper lacks an ablation study to explore different strategies of layer selection, such as a random selection baseline.

---

### Official Review · Reviewer_AfsH · 2025-10-25

**Soundness:** 2
**Presentation:** 2
**Contribution:** 3
**Rating:** 4
**Confidence:** 4

**Summary:**

The paper proposes a Noise-Guided Layer Selection (NGLS) method. It injects Gaussian noise into auxiliary data and measures inter-layer similarity to select backbone layers that contribute most to domain invariance. The approach is novel, simple, and effective, enabling parameter-efficient domain adaptive object detection. Experiments are conducted on both SFOD and UDAOD tasks. The results show that the method greatly reduces the number of trainable parameters while maintaining or even surpassing baseline performance, demonstrating strong practical value.

**Strengths:**

－ The paper is the first to apply noise-guided layer selection to domain adaptive object detection. By combining layer similarity analysis, it introduces a lightweight and pluggable selection mechanism.

－The method shows remarkable parameter efficiency, achieving a significant reduction in trainable parameters across multiple baselines.

－ The approach is simple and reproducible, relying only on an auxiliary dataset (COCO) and cosine similarity, without depending on source or target domain labels.

**Weaknesses:**

－ As noted in Paper 062, the proposed NGLS is inspired by the works of Meng and Zhang. However, its design appears relatively simple and seems to merge ideas from both. The authors are encouraged to conduct deeper analysis to better emphasize the novelty and include direct comparisons, especially with Zhang’s method, to demonstrate the effectiveness and unique advantages of NGLS.

－The idea of identifying robust feature layers by comparing clean and noise-injected images is reasonable. However, it remains unclear whether these layers maintain or improve their robustness after target-domain optimization. The authors are encouraged to include further theoretical discussion and visualization-based analysis to clarify this mechanism.

－ It is suggested to include additional experiments with randomly selected layers to verify the necessity and robustness of the NGLS strategy.

－ More noise analysis is encouraged, such as examining the impact of different Gaussian parameters and various noise distributions on layer selection results.

－ There are also minor spelling errors, for example, “adaptaiton” on line 108 should be corrected.

**Questions:**

－ Clarify novelty and provide comparative analysis of NGLS.

－  Clarify robustness of feature layers after target-domain optimization.

－  More experimental analysis.

For details, please refer to the Weaknesses section.

---

### Official Review · Reviewer_xcWJ · 2025-10-31

**Soundness:** 2
**Presentation:** 2
**Contribution:** 2
**Rating:** 4
**Confidence:** 3

**Summary:**

This paper addresses the parameter efficiency issue in Domain Adaptive Object Detection (DAOD) and proposes a novel method called NGLS (Noise-Guided Layer Selection). Its core idea is to inject Gaussian noise into auxiliary data and measure the robustness of each layer in the backbone network to the noise, identifying and fine-tuning only the layers that are most critical to domain invariance.

**Strengths:**

1. This paper proposes a novel plug-and-play layer selection method, NGLS, that leverages Gaussian
noise perturbation to identify backbone layers most robust to domain shifts.
2. By fine-tuning only the NGLS-selected layers, our approach substantially reduces the number of parameters while achieving comparable or superior performance to full-model finetuning in both SFOD and UDAOD settings.
3. NGLS only requires a handful of unlabeled auxiliary images for robust layer selection,
eliminating the need for source or target domain labels and supporting practical, dataefficient adaptation.

**Weaknesses:**

1. Lack of in-depth discussion regarding the relationship between "noise" and "domain invariance", which is a critical issue. Why should layers that are robust to Gaussian noise necessarily be robust to domain shift?
2. Limitation of the layer selection criterion. NGLS selects layers based solely on the output similarity of individual layers, ignoring the collaborative interactions among layers. It is possible that certain layers appear sensitive to noise when considered in isolation, but become robust to domain shift when combined with other layers. NGLS would erroneously exclude such layers.
3. The validation of generalizability across different backbone architectures is insufficient. The experiments are primarily conducted on ResNet. It remains unclear whether the method remains effective for more modern backbones (e.g., Swin Transformer), as their layer structures and feature extraction mechanisms fundamentally differ from those of CNNs.

**Questions:**

Please refer to the Weakness.

---

### Official Review · Reviewer_X8kH · 2025-11-01

**Soundness:** 3
**Presentation:** 3
**Contribution:** 2
**Rating:** 2
**Confidence:** 4

**Summary:**

The paper proposes a method to identify layers that are invariant to perturbations in their input, which are then adapted to previously unseen domains by domain-adaptive object detection methods. Invariance is assessed on an auxiliary dataset by perturbing inputs to each layer with gaussian noise and measuring cosine similarity to the unperturbed features. By keeping non-invariant layers frozen, the number of trainable layers and GPU memory usage is reduced, while largely keeping performance on the target domain intact, compared to full adaptation.

**Strengths:**

The proposed method is largely complementary to existing DAOD methods. The method for finding invariant layers is simple but effective. The experiments and ablations are thorough.

**Weaknesses:**

I have two primary concerns. First, I think the need for reducing the number of trainable parameters could be more strongly motivated. I can see a hypothetical need for real-world edge device deployments, but how much of a bottleneck  the number of trainable parameters would be in that case is unclear to me. Second, I feel like the paper does not intuitively motivate and discuss its main contribution very well. The process described in the paper “select[s] layers that maintain high similarity for adaptation.”. Intuitively, those are the layers most invariant to domain shift, and as such should see the _least_ need to be fine-tuned, compared with layers that are not invariant under domain shift.

**Questions:**

* Why does it make sense to inject noise before each layer? Why not just inject noise / perform some other type of augmentation on the input images?
* Intuitively, why is there need to tune the invariant layers, which should need little to no adaptation? Table 9 shows experimental evidence, but misses an intuitive explanation for this unexpected phenomenon.

---

### Official Review · Reviewer_fF8Z · 2025-11-02

**Soundness:** 2
**Presentation:** 2
**Contribution:** 2
**Rating:** 4
**Confidence:** 4

**Summary:**

This paper proposes Noise-Guided Layer Selection (NGLS), a plug-and-play method for parameter-efficient domain adaptive object detection. Instead of fine-tuning all model parameters, NGLS selectively fine-tunes only backbone layers that are most robust to domain shifts. It identifies these layers by injecting Gaussian noise into an auxiliary dataset and measuring layer-wise cosine similarity between clean and noisy features.

**Strengths:**

1. The paper is easy to follow and clearly written, with well-structured explanations that make the methodology understandable.
2. The proposed approach is conceptually simple and methodologically concise.
3. The algorithm is designed in a plug-and-play manner, demonstrating strong generalizability and potential for seamless integration into various DA frameworks.

**Weaknesses:**

1. I have some concerns, that is the using auxiliary datasets to identify layers whose outputs remain similar under Gaussian noise perturbation does not necessarily indicate that these layers have learned domain-invariant representations. Rather, it only shows that these layers are robust to the injected Gaussian noise. However, robustness to Gaussian noise cannot be directly equated with robustness to domain shift, as the two represent fundamentally different types of distributional variations.
2. A minor concern is that DA aims to train model parameters to extract domain-invariant representations. Intuitively, this process should involve optimizing the more domain-specific layers, since they are most sensitive to distributional differences. It is therefore somewhat counter-intuitive that the authors choose to fine-tune layers that are less affected by perturbations. The paper would benefit from a clearer justification explaining why adapting the more stable layers leads to better cross-domain generalization.

**Questions:**

1. While the paper reports a substantial reduction in the number of trainable parameters, the improvement in GPU memory usage is relatively modest (only around 20%). Given that most DAOD and SFOD methods are not highly memory-intensive to begin with, it would be helpful if the authors could provide a detailed analysis of training efficiency in terms of time or computational cost, possibly with a comparison table.
2. In the ablation study, an experiment where layers are randomly sampled for fine-tuning may be included, to compare against the proposed noise-guided selection. Additionally, when comparing low-similarity and high-similarity layers, the number of trainable parameters should be kept consistent, rather than determined by manually set thresholds.

---

### Note · Authors · 2025-11-14

**Comment:**

We thank all reviewers for their time and constructive feedback. After careful consideration, we have decided to withdraw this paper.

**Withdrawal Confirmation:**

I have read and agree with the venue's withdrawal policy on behalf of myself and my co-authors.